# Resource

# A large-scale cancer-specific protein–DNA interaction network

Yunwei Lu[1],*, Anna Berenson[1,2],*, Ryan Lane[1],*, Isabelle Guelin[1], Zhaorong Li[3], Yilin Chen[1], Sakshi Shah[1], Meimei Yin[1], Luis Fernando Soto-Ugaldi[4], Ana Fiszbein[1,2,3], Juan Ignacio Fuxman Bass[1,2,3]

Cancer development and progression are generally associated with gene dysregulation, often resulting from changes in the transcription factor (TF) sequence or expression. Identifying key TFs involved in cancer gene regulation provides a framework for potential new therapeutics. This study presents a large-scale cancer gene TF-DNA interaction network, as well as an extensive promoter clone resource for future studies. Highly connected TFs bind to promoters of genes associated with either good or poor cancer prognosis, suggesting that strategies aimed at shifting gene expression balance between these two prognostic groups may be inherently complex. However, we identified potential for oncogene-targeted therapeutics, with half of the tested oncogenes being potentially repressed by influencing specific activators or bifunctional TFs. Finally, we investigate the role of intrinsically disordered regions within the key cancer-related TF ESR1 in DNA binding and transcriptional activity, and found that these regions can have complex trade-offs in TF function. Altogether, our study broadens our knowledge of the TFs involved in cancer gene regulation and provides a valuable resource for future studies and therapeutics.

## Introduction

Gene expression is often dysregulated in cancer because of changes in copy number, mutation or epigenetic changes in promoter and enhancer regions, or changes in the expression or activity of transcription factors (TFs) and chromatin-modifying enzymes (Djakiew, 2000; Vervoort et al, 2022). Among the affected genes are those involved in cell differentiation, proliferation, apoptosis, DNA repair, immune regulation, and general biological processes such as translation and RNA processing, ultimately contributing to cancer development, progression, and metastasis (Hanahan & Weinberg, 2011; Carrasco et al, 2023).

The higher expression of certain genes has been associated with good or poor cancer prognosis (Sjostedt et al, 2020). Some of these genes are associated with prognosis only in specific cancers, whereas others have the same or opposing associations in different cancer types. For instance, the elevated expression of *GNAS* has been found to promote cell proliferation in breast cancer (Jin et al, 2019), whereas the reduced expression of *CAMTA1* has been linked to adverse outcomes in neuroblastoma patients (Henrich et al, 2006). Therefore, a promising potential cancer therapeutic strategy could consist of shifting the balance in expression between poor and good prognosis genes, which may eventually lead to increased cancer survival. The rational design of this strategy involves identifying TFs that preferentially regulate the expression of either poor or good prognosis genes. This requires the delineation of large-scale gene regulatory networks that evaluate the binding of hundreds of TFs to the regulatory elements of cancer-related genes.

Multiple experimental methods have been developed to identify TF-DNA interactions. Chromatin immunoprecipitation followed by sequencing (ChIP-seq) and CUT&RUN are widely used to identify the genomic DNA regions that a TF binds in vivo. Although these methods have provided extensive datasets, in particular by large consortia such as the ENCODE Project, lowly expressed TFs and TFs for which ChIP-grade antibodies are not available remain understudied (Luo et al, 2020). Enhanced yeast one-hybrid (eY1H) assays provide a high-throughput complementary gene-centered method to identify the repertoire TFs that bind to DNA regions of interest by testing >1,000 TFs simultaneously (Reece-Hoyes et al, 2011b; Fuxman Bass et al, 2015; Fuxman Bass et al, 2016a; Berenson & Fuxman Bass, 2023). In eY1H assays, each TF is fused to the yeast Gal4 activation domain and expressed in a separate yeast strain; binding of the TF to a DNA region of interest induces the expression of two reporter genes, *HIS3* and *LacZ*, allowing yeast to grow and turn blue on readout plates. Given that eY1H assays involve expressing exogenous human TFs in yeast, they can detect interactions involving TFs that have low endogenous expression or that lack suitable antibodies. Although eY1H assays cannot test binding

[1]Biology Department, Boston University, Boston, MA, USA   [2]Molecular Biology, Cellular Biology and Biochemistry Program, Boston University, Boston, MA, USA   [3]Bioinformatics Program, Boston University, Boston, MA, USA   [4]Tri-Institutional Program in Computational Biology and Medicine, New York, NY, USA

Correspondence: fuxman@bu.edu
*Yunwei Lu, Anna Berenson, and Ryan Lane contributed equally to this work

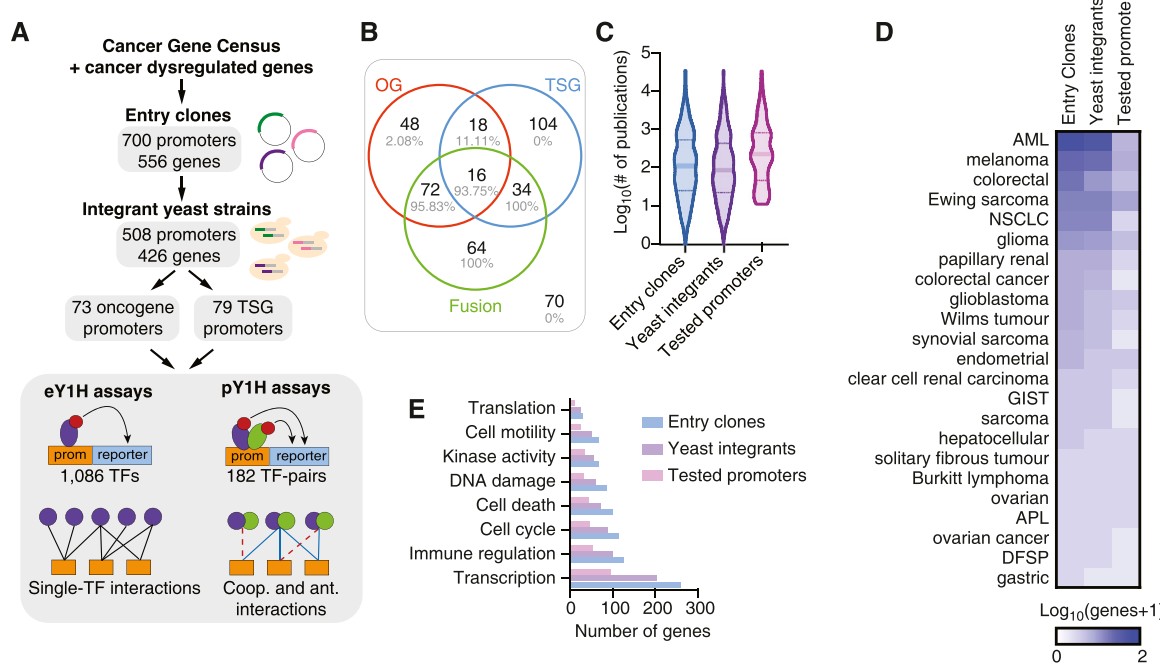

**Figure 1. Generation of clone and yeast resource for cancer gene promoters.**
**(A)** Schematic of the Gateway-compatible cancer gene promoter resource. Cancer genes were selected from the Cancer Gene Census, as well as genes dysregulated in cancer. An entry clone resource of 700 promoters (556 genes) was generated, as well as a yeast integrant resource corresponding to 508 promoters (426 genes). This yeast resource was tested in eY1H and pY1H assays for TF-DNA interactions. **(B)** Venn diagram of the number of oncogenes (OG), tumor suppressor genes (TSG), and genes involved in fusions for which yeast integrants were generated. The percentage of genes related to translocation in each group is marked in gray. **(C)** Violin plots correspond to the distribution of the number of publications per gene included in the entry clone resource, the yeast integrant collection, and the yeast integrants tested by eY1H/pY1H assays. **(D)** Number of genes associated with different cancer types among those in the set of entry clones, yeast integrants, and tested by eY1H assays. **(E)** Number of genes associated with different biological functions for genes included in the entry clone resource, the yeast integrant collection, and the yeast integrants tested by eY1H/pY1H assays.

of heterodimeric TFs, we have recently addressed this limitation by developing paired yeast one-hybrid (pY1H) assays. This method evaluates pairs of TFs to detect cooperative binding and antagonism at DNA regions of interest (Berenson et al, 2023).

In this study, we generated a clone resource of 700 cancer-related gene promoters and used both eY1H and pY1H platforms to examine binding of monomeric/homodimeric and heterodimeric TFs to the promoters of 136 cancer-related genes. We identified 1,350 interactions between 265 TFs and the promoters of 108 cancer genes, and leveraged our promoter library to study disordered regions in the breast cancer–related TF estrogen receptor α (ESR1). Overall, our work provides new insights into the study of the regulation of cancer genes and provides a clone and data resource for the scientific community.

## Results

### Generation of a comprehensive clone resource of cancer gene promoters

Systematic studies of TF-DNA binding and transcriptional activity often require large-scale clone resources of regulatory DNA elements—such as promoters and enhancers—that can be tested

across functional assays. Considering that promoters are the primary drivers of gene expression, whereas enhancers require more specific spatiotemporal contexts to function effectively, we focused on promoter regions of cancer-related genes to study their regulation by TFs (Bergman et al, 2022). We initially selected 697 genes from the Cancer Gene Census (Tate et al, 2019), as well as 114 additional genes whose expression is often dysregulated in cancer. For 190 of these genes, we have also included alternative promoters, in cases where H3K27ac or H3K4me3 marks were observed in data from the ENCODE Project (ENCODE Project Consortium, 2012). We successfully cloned 700 promoter sequences—each comprising 2 kb of sequence immediately upstream of a transcription start site—corresponding to 556 cancer-related genes, generating a Gateway-compatible resource for easy transfer into different destination vectors that can be used in a variety of functional assays (e.g., eY1H, pY1H, and luciferase assays) (Fig 1A, Table S1).

To perform TF-DNA binding studies using eY1H and pY1H assays, we also transferred these clones into the appropriate destination vectors upstream of two reporter genes (*HIS3* and *LacZ*) and successfully generated integrant yeast strains for 508 promoters corresponding to 426 cancer-related genes. Among these, 358 genes were classified as oncogenes, tumor suppressor genes, or genes involved in fusions, with a similar number of genes in each of these classes (Fig 1B), whereas the remaining 68 genes are not classified into these categories by the Cancer Gene Census. Our set

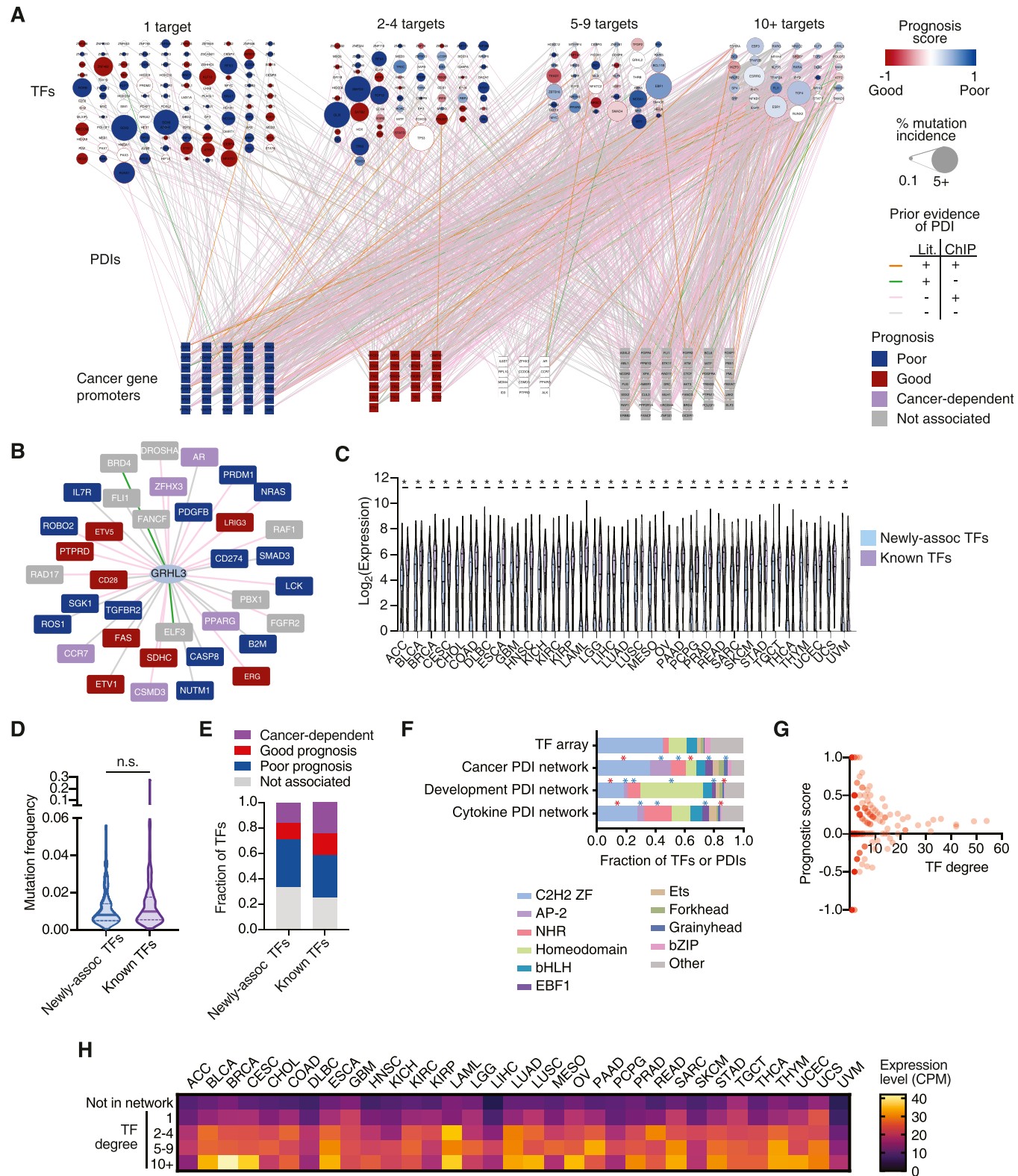

**Figure 2. Large-scale cancer TF-DNA interaction network.**
**(A)** Cancer TF-DNA interaction network determined using eY1H and pY1H assays. Circular nodes represent TFs, whereas squares represent cancer gene promoters. Interactions are represented by edges colored based on whether there is evidence by ChIP-seq (pink), literature (green), both (orange), or neither (gray). TF nodes are colored based on the prognostic score calculated as (#poor prognosis targets - #good prognosis targets)/(# total number of targets). The borders of TF nodes are colored based on whether the TF is listed (red) in CGC. TF node size indicates the % of non-synonymous mutations across all cancers. Cancer gene promoters are colored based

of genes included both highly studied genes, with >1,000 publications in PubMed, and lowly studied genes with <10 publications, including genes with known associations with a variety of cancer types (Fig 1C and D). Regarding biological functions, our clone resource includes genes associated with transcription, immune regulation, cell cycle, cell death, DNA damage, and other cancer-related functions (Fig 1E). We did not observe any major bias between genes for which entry clones or yeast strains were successfully generated (Fig 1C–E).

### A comprehensive cancer-associated TF-DNA network

The abnormal expression of cancer-related genes can lead to oncogenesis, cancer progression, and metastasis (Li et al, 2020). Dysregulation of these genes can be caused by increased or decreased binding of certain TFs as a result of changes in TF expression, by mutations in TFs or TF binding sites, or by alteration in TF activity because of dysregulation of upstream signaling pathways (Gonda & Ramsay, 2015). To identify the TFs that bind to the promoters of cancer-related genes, we used eY1H and pY1H assays, which can identify the binding of hundreds of single or pairs of TFs to DNA elements of interest in parallel. We prioritized 136 genes (152 promoter sequences), corresponding to 66 oncogenes and 70 tumor suppressor genes, including genes that were highly studied in the literature and promoters with high H3K27ac indicative of active usage as a regulatory DNA region (Fig 1C). These genes had a similar representation of biological process gene ontologies relative to the rest of the gene promoter clone resource ($P > 0.05$ by a hypergeometric test) (Fig 1E), suggesting that our prioritized subset was functionally unbiased. We tested these DNA sequences against 1,086 TFs using eY1H assays (147,696 TF-DNA pairs tested in quadruplicate). In addition, we tested 123 of these promoter sequences (selected based on low levels of autoactivity seen in eY1H) against a collection of 182 TF pairs and corresponding monomers using pY1H assays (22,386 TF-TF-DNA sets tested in quadruplicate). In total, we detected 1,350 TF-DNA interactions between 265 individual TFs (including 30 heterodimeric TFs) and the promoters of 108 genes (Fig 2A, Table S2). Of these TFs, 84 are classified as transcriptional activators, 33 as repressors, and 42 as bifunctional TFs based on their annotated effector domains (Soto et al, 2022).

Among the interactions detected by eY1H and pY1H assays, we found 543 interactions that were previously identified in ChIP-seq experiments, 21 interactions that were reported in the literature, and 61 interactions that were reported in both (Fig 2A). Furthermore, consistent with previous studies, we found that TFs whose interactions did not present evidence by ChIP-seq were assayed less frequently than TFs for which ChIP-seq evidence was found (Fuxman Bass et al, 2015) (Fig S1A). This suggests that further ChIP-

seq datasets are likely to add evidence for the interactions detected by eY1H and pY1H assays, and illustrates the high quality of our cancer TF-DNA network. Importantly, we found 725 novel interactions, showing that our network also expands from previously reported interactions. This includes novel interactions involving TFs already known to bind to the promoters of some of the cancer genes tested. For example, GRHL3, a TF known to stimulate migration of endothelial cells and previously linked to different types of cancers (Wang et al, 2017), had 19 ChIP-seq and two literature interactions with our set of 152 promoters. Here, we found 15 additional interactions using eY1H assays (Fig 2B). This set of 36 genes displayed significant enrichment in the cell differentiation gene ontology term, which is consistent with previous studies showing that GRHL3 is crucial for inducing genes within the epidermal differentiation complex, which supports terminal differentiation, suppressing hyperproliferation (Lin et al, 2020; Huang et al, 2022). We also found interactions involving TFs that were previously not known to regulate any of the genes in our network. For example, ESRRG and CEBPE interact with 15 and 5 cancer gene promoters, respectively, but interactions with our set of promoters were not reported in the literature nor in ChIP-seq experiments. Both TFs are known to have roles in different cancers such as myeloid leukemia (CEBPE) and gastric cancer and retinoblastoma (ESRRG) (Heckler et al, 2014; Kang et al, 2018; Li et al, 2019).

Although these "newly associated" TFs had overall lower expression levels across cancers than TFs known to regulate this set of cancer genes (Fig 2C), there is a significant overlap between both distributions, with 35% of newly associated TFs being expressed at more than 100 CPM in at least one cancer type. More importantly, the set of newly associated TFs has a similar mutation rate in cancer and a similar likelihood of having a significantly poor or good association with prognosis than TFs already known to regulate this set of cancer genes (Fig 2D and E). Altogether, this shows that the eY1H and pY1H interaction data capture TFs relevant to cancer and provides direct binding evidence supporting existing literature and ChIP-seq data, while identifying many novel interactions to delineate a more comprehensive cancer-related TF-DNA interaction network.

### TF family representation in the cancer-associated TF-DNA network

We observed interactions involving all major TF families including homeodomains, Cys2His2 zinc fingers (ZF-C2H2), nuclear hormone receptors (NHRs), basic helix–loop–helix (bHLH), and basic leucine zippers (bZIP). Compared with the proportion of TF families in the array, we found an over-representation of interactions involving the EBF1, grainyhead, NHR, and AP-2 families (Fig 2F), which are

on whether their expression is associated with poor (blue), good (red), or cancer-dependent prognosis (white). **(B)** Interaction network involving GRHL3. **(C)** Expression levels across cancer types for TFs known to bind/regulate our set of cancer genes (purple), and newly associated TFs (blue). Statistical significance was determined by two-tailed Mann–Whitney's $U$ test. **(D)** Violin plot depicting the mutation frequency across cancers for TFs known to bind/regulate our set of cancer genes (purple), and newly associated TFs (blue). Statistical significance was determined by two-tailed Mann–Whitney's $U$ test. **(E)** Fraction of TFs whose expression levels are associated with poor or good cancer prognosis for TFs known to bind/regulate our set of cancer genes, and newly associated TFs. **(F)** Fraction of TFs and TF-DNA interactions corresponding to different TF families in the TF array and the cancer promoter, cytokine promoter, and developmental enhancer networks. *$P < 0.05$ by a proportion comparison test. Blue and red asterisks represent enrichment and depletion relative to the TF array, respectively. **(G)** Prognosis scores for TFs with different numbers of targets (degree) in the cancer TF-DNA interaction network. **(H)** Median expression levels in each cancer type for TFs absent from the network and TFs with different degrees.

known to play important roles in tumor growth and progression via diverse mechanisms (Tsigelny et al, 2014; Zhang et al, 2020; Hu et al, 2021; Carrasco et al, 2023). Interestingly, we found that AP-2, in particular TFAP2B, is also enriched compared with previous screens against developmental enhancers and cytokine gene promoters (Figs 2F and S1B and C), suggesting that this TF family may be more actively involved in cancer regulation. Indeed, AP-2 family members such as TFAP2A, TFAP2B, and TFAP2C have been shown to be involved in different cancer types such as glioblastoma, melanoma, acute myeloid leukemia, pancreatic cancer, and colorectal cancer (Kolat et al, 2019; Raap et al, 2021).

Conversely, we observed depletion of interactions involving homeodomain TFs, which were previously found to be enriched in the developmental enhancer network. This is consistent with their roles in the development of anatomical features during early embryogenesis (Banerjee-Basu & Baxevanis, 2001), but less so in cancer. This illustrates that although cancer is inherently a developmental/differentiation process, the underlying gene regulatory networks use different sets of TFs compared to developmental networks.

### Highly connected TFs bind to the promoters of good or poor prognosis genes

TFs in the cancer network are generally more highly expressed across cancers than TFs for which we did not detect any interactions (Fig 2H). Furthermore, TFs in the network also tend to be more often differentially expressed in tumor samples versus matched normal controls (Fig S2). Together, this suggests that TFs in the cancer TF–DNA network are cancer-relevant.

TFs in the network bind to a widely different number of promoters, ranging from 1 to 54 promoters. Half (117/265) of TFs bind to just one promoter, whereas 13.58% (36/265) bind to 10 or more (Fig 2A). This is consistent with a power-law distribution, which is frequently observed in gene regulatory networks (Fig S1D) (Lima-Mendez & van Helden, 2009). TFs absent from the network and those with 1 or few interactions have overall lower expression levels across cancer types than highly connected TFs (TF hubs) (Fig 2H). This suggests that TF hubs may have a substantial impact on gene regulatory networks and represent potentially valuable drug targets as they coordinate the expression of multiple genes. However, we found that none of the TFs with 10 or more interactions in our network display any significant bias toward binding to the promoters of genes associated with either good or poor prognosis (Figs 2G and S3). For example, EGR1, a TF that binds to 27 genes in our network, 7 known to be associated with poor and 4 associated with good cancer prognosis, has a complex role in cancer with both tumor-suppressing and tumor-promoting activities (Adamson & Mercola, 2002; Wang et al, 2021). Similarly, RUNX2, another TF hub that has a dual transcriptional role (i.e., can act as an activator or a repressor), bound both to the promoters of genes associated with poor and good cancer prognosis. This complex and diverse set of binding targets associated with TF hubs suggests that targeting them could be challenging as the overall rewiring of the cancer gene regulatory network may be hard to anticipate. Indeed, TF hubs in our network do not have a higher mutation rate in cancer compared with moderately connected TFs, consistent with the lack of bias in binding to good or poor prognosis genes (Fig 2A).

### Identification of TFs as potential targets to reduced oncogene expression

During cancer development and progression, oncogenes may undergo mutations, amplifications, or structural changes, leading to poor patient outcomes. After decades of efforts, drugs such as AMG510 (sotorasib) that targets KRAS have been developed to inhibit oncogene activity (Huang et al, 2021); however, numerous oncogenes still resist direct targeting. A potential alternative therapeutic approach involves reducing the expression of the oncogene using knockdowns or by modulating gene transcription by targeting TF activity. A barrier to this latter approach is the potential for concomitant up-regulation of poor prognosis genes or the down-regulation of good prognosis genes. To nominate TF candidates that could be targeted (e.g., by down-regulation, targeted degradation, or small molecule inhibition) to decrease the expression of each oncogene tested, we established the following criteria: (1) the TF is an activator (or bifunctional), and (2) it preferentially binds to the promoters of poor prognosis genes (prognosis score > 0.33). For 25 of the 51 oncogenes tested, we found at least one TF that meets the above criteria (Fig 3). This includes druggable TFs such as PPARG and RARA, as well as TFs for which drugs have not yet been developed. For instance, in our network, six TFs with an activation domain bound to the promoter of NUTM1, which are frequently rearranged or fused with other genes and whose overexpression is associated with poor prognosis in patients with NUT carcinoma (French 2018), B-cell precursor acute lymphoblastic leukemia (Hormann et al, 2019), oral squamous cell carcinoma (OSCC) (Riaz & Khan 2023 Preprint), and thyroid carcinoma (Allison et al, 2022). Treatments, especially for NUT carcinoma, have been undergoing development during the last decades; however, these drugs mostly target genes fused to NUTM1 in cancer; for example, BET inhibitors target BRD4 in the BRD4-NUTM1 fusion protein. Our findings suggest that we could potentially target TF activators such as ARNTL and SRF to tune down the expression of NUTM1.

### Cooperative and antagonistic TF binding interactions with cancer gene promoters

In addition to the regulation by monomeric and homodimeric TFs, the expression of cancer-associated genes can also be modulated by higher order functional relationships between TFs at promoter sequences, including cooperativity and antagonism between TF pairs (Jolma et al, 2015; Morgunova & Taipale, 2017; Ibarra et al, 2020; Hu et al, 2022). TFs from a variety of families are known to bind DNA cooperatively as heterodimers, including NF-κB (Oeckinghaus & Ghosh, 2009), AP-1 (Karin et al, 1997), STATs (Lim & Cao, 2006; Delgoffe & Vignali, 2013), nuclear receptors, and other bZIP (Rodriguez-Martinez et al, 2017) and bHLH TFs (de Martin et al, 2021). However, the extent to which cooperativity between TFs is involved in cancer gene targeting has not been systematically explored. Furthermore, DNA binding antagonism between TFs, in which dimerization prevents binding of a TF to certain DNA targets, has not typically been considered as a widespread transcriptional regulatory mechanism.

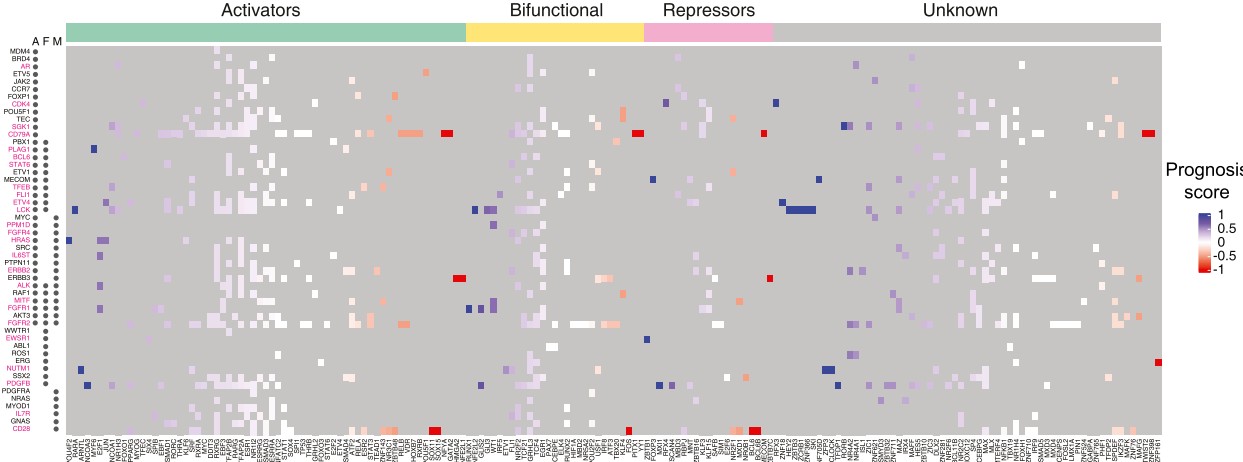

**Figure 3. Heatmap of potentially targetable TFs to reduce oncogene expression.**
Heatmap of prognostic scores for TFs that bind to oncogene promoters. TFs are classified as potential activators, bifunctional, or repressors based on annotated effector domains in TFRegDB. Oncogenes that contribute to cancer development through amplifications (A), fusions (F), or mutations (M) are indicated next to the gene name. Oncogenes indicated in magenta have at least one TF that is activator/bifunctional with a prognosis score > 0.33.

To obtain a deeper understanding of how these relationships can contribute to specificity of cancer gene regulation, we used pY1H assays, which test the binding of pairs of TFs to DNA regions of interest (Berenson et al, 2023). We evaluated the binding of 182 TF pairs to 123 cancer gene promoters and detected 90 cooperative and 136 antagonistic interactions, involving a total of 66 promoters and 67 TF pairs (Fig 4A, Table S3). We found 25 TFs to exclusively participate in cooperative binding events, 27 TFs exclusively participated in antagonistic binding events, and 21 TFs were observed to be involved in both cooperative and antagonistic interactions, depending on the TF partner or the gene promoter (Fig 4B). Of the 67 TF pairs that showed at least one type of pY1H interactions, 23 exclusively participated in cooperativity (e.g., RXRG-NR1H3, SPDEF-ATF2, and TCF21-TCF4) and 37 of them exclusively participated in antagonism (e.g., MAX-MNT, DLX2-MLXIP, and HIF1A-RUNX2) (Fig 4A and C). Interestingly, 7 TF pairs participated in both kinds of interactions (e.g., MAX-MYC and DLX2-ZNF281). Altogether, this suggests that individual TFs may regulate different target genes depending on their interacting TF partners.

We observed that TF pairs from various families, including both intra- and inter-family pairs, exhibited cooperative and antagonistic binding (Fig 4A). We also noted that TFs from different families tend to show a preference for specific types of functional interactions (Fig 4D). For instance, NHRs predominantly engage in cooperative interactions, consistent with the well-known heterodimeric partnership in this TF family. For example, we observed that ESR1 and NR2F2, two TFs that are highly mutated in various types of cancer, cooperatively bind to 10 different promoters in our network. bHLHs are often involved in antagonistic relationships. This is mostly driven by the cancer-related TF MAX whose binding is antagonized by many other bHLH TFs such as MNT, MXD1, and MXD4.

For 21 TFs, we observed different functional interactions depending on the TF partner. For example, DLX2, a TF known to be up-regulated during epithelial–mesenchymal transition and to promote

cell survival (Tiwari et al, 2012), exhibited diverse binding relationships across TF partners at various DNA sequences. In our screen, DLX2 independently bound to the promoters of 11 genes, and at 5 of them, it showed no functional interaction with any partner TF (Fig 4E). However, at 5 promoter sequences, DLX was antagonized by other TFs such as MLXIP, MSX1, and MXD1. Interestingly, DLX2 showed cooperativity or was antagonized by ZNF281 and MLX depending on the promoter sequence (Fig 4E). We had a similar observation for MYC, an essential regulator of cell growth overexpressed in many tumors, known to heterodimerize with MAX and cooperatively bind to DNA. We observed this cooperation between MYC and MAX at three promoters; however, we also noted that MYC was antagonized by MAX at the *PDGFB* promoter, known to be targeted by MYC (Winkler et al, 2021). This aligns with a previous report indicating that MAX could antagonize MYC in a dose-dependent manner through the competition of MAX-MAX and MYC-MAX dimers for their common target DNA sites (Amati et al, 1993; Amati & Land, 1994). Altogether, this highlights the complexity of higher order TF binding, which can be heavily influenced not only by the partners of a TF but also by the target sequences involved, as we have previously observed for a small set of cytokine genes (Berenson et al, 2023).

## Intrinsically disordered regions affect ESR1 binding to DNA

In addition to broadening our knowledge of which TFs participate in the cancer gene regulatory network, we were also interested in conducting a more in-depth study of key cancer-related TFs. Estrogen receptor α (ESR1) is a TF that is frequently mutated or up-regulated in breast cancer, and is therefore an important candidate for further functional study. ESR1 contains two structured domains—the DNA binding and the ligand binding domains—flanked by three intrinsically disordered regions (IDRs): the N-terminal region (amino acids 1–180) containing the transactivation function-1 (AF-1) domain, the hinge region (amino acids 254–305) connecting the DNA binding and ligand binding domains, and the C-terminal region (amino acids 553–595)

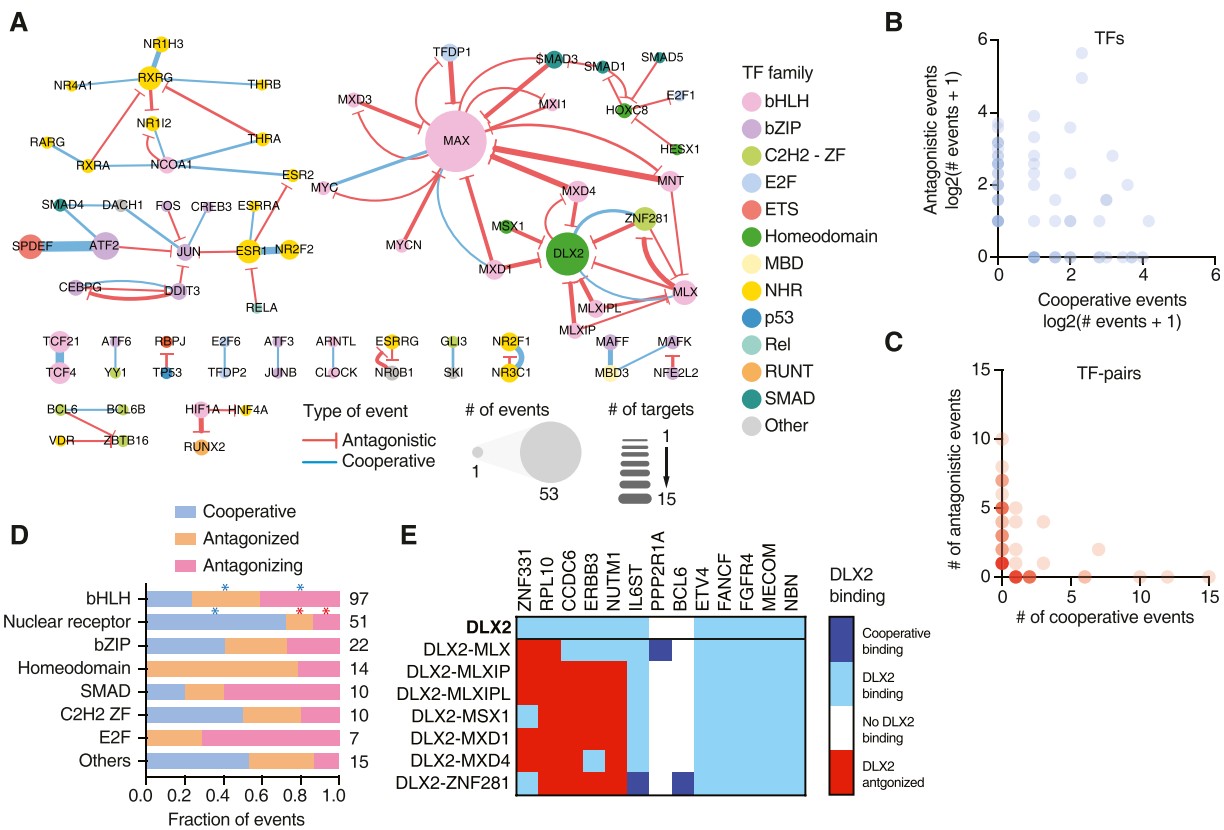

**Figure 4. TF binding cooperativity and antagonism at cancer gene promoters.**
**(A)** Network of cooperative (blue) and antagonistic (red) relationships between TFs at the cancer gene promoters screened. Node size indicates the number of binding events for that TF. Edge width represents the number of cooperative or antagonistic events involving a specific TF pair. **(B)** Number of cooperative and antagonistic events observed for individual TFs. **(C)** Number of cooperative and antagonistic events observed for TF pairs. **(D)** Fraction of events where a TF binds cooperatively, is antagonized by another TF, or antagonizes the binding of another TF for each TF family. *$P < 0.05$ by Fisher's exact test. Blue and red asterisks represent enrichment and depletion, respectively. The number of binding events for that TF family is listed on the right. **(E)** Heatmap of interactions involving DLX2, either on its own or together with other TF partners at 13 cancer gene promoters. Dark blue—cooperative binding; light blue—DLX2 binding not influenced by partner TF; white—no DLX2 binding; and red—DLX2 binding antagonized by TF partner.

(Fig 5A). IDRs in TFs have long been associated with roles in transcriptional activity, including most reported activation and repression domains (Soto et al, 2022). Recently, ChIP-seq studies have shown that IDRs can also modulate DNA binding across the genome, likely by affecting protein–protein interactions with other TFs and cofactors or by mediating condensate formation (Brodsky et al, 2021). We hypothesized that IDRs may also modulate DNA binding in a heterologous context, in the absence of other TFs and cofactors of the same species. We therefore set out to identify the contributions of ESR1 IDRs to both DNA binding and transcriptional activity and to identify how known mutations in ESR1 IDRs might disrupt these functions.

To determine whether these IDRs affect DNA binding, we performed eY1H screens using wild-type ESR1, three truncations of the N-terminal IDR (ΔN59, ΔN119, and ΔN179), two truncations of the C-terminal IDR (ΔC23 and ΔC43), a replacement of the hinge region with 23 tandem copies of Gly-Ser (Hinge[GS]$_{23}$) to maintain flexibility of the linker region while removing the endogenous sequence, and 11 cancer-associated mutations in these IDRs reported in COSMIC (Fig 5A, Table S4). Our entire collection of cancer promoters was tested against each ESR1 construct in the presence or absence of 100 nM estradiol. We found that estradiol was generally needed for DNA binding activity, consistent with the need for estradiol-

mediated dimerization for ESR1 binding to estrogen response elements (Yasar et al, 2017) (Fig 5B). Each interaction tested was manually scored on a scale from 0 (no reporter signal) to 5 (very strong reporter signal). We observed that progressive N-terminal truncations led to increased reporter signal for gene promoters that already bound the full-length wild-type ESR1 (type 1 promoters) and even led to novel DNA binding events (type 2 promoters) (Fig 5C, Table S5). This suggests that the N-terminal IDR, in particular amino acids 1–119, suppresses ESR1 binding to DNA. Truncations of the C-terminal region had a less clear effect, with a 23 amino acid truncation mildly increasing or decreasing DNA binding depending on the promoter sequence, while the 43 amino acid truncation mildly reducing DNA binding strength. Interestingly, we found that replacing the hinge region for a (GS)$_{23}$ flexible linker led to a strong reduction in DNA binding strength and the number of promoter sequences bound, suggesting that the hinge region is necessary for proper DNA binding. We confirmed the reduced binding of the Hinge(GS)$_{23}$ ESR1 construct to the *AFF2* and *NBL1* promoters using reporter-based protein–DNA interaction assays in HEK293T cells treated with 100 nM estradiol (Fig 5D and E).

Most (10/11) cancer mutations tested did not affect ESR1 binding to the cancer promoters tested, suggesting that these

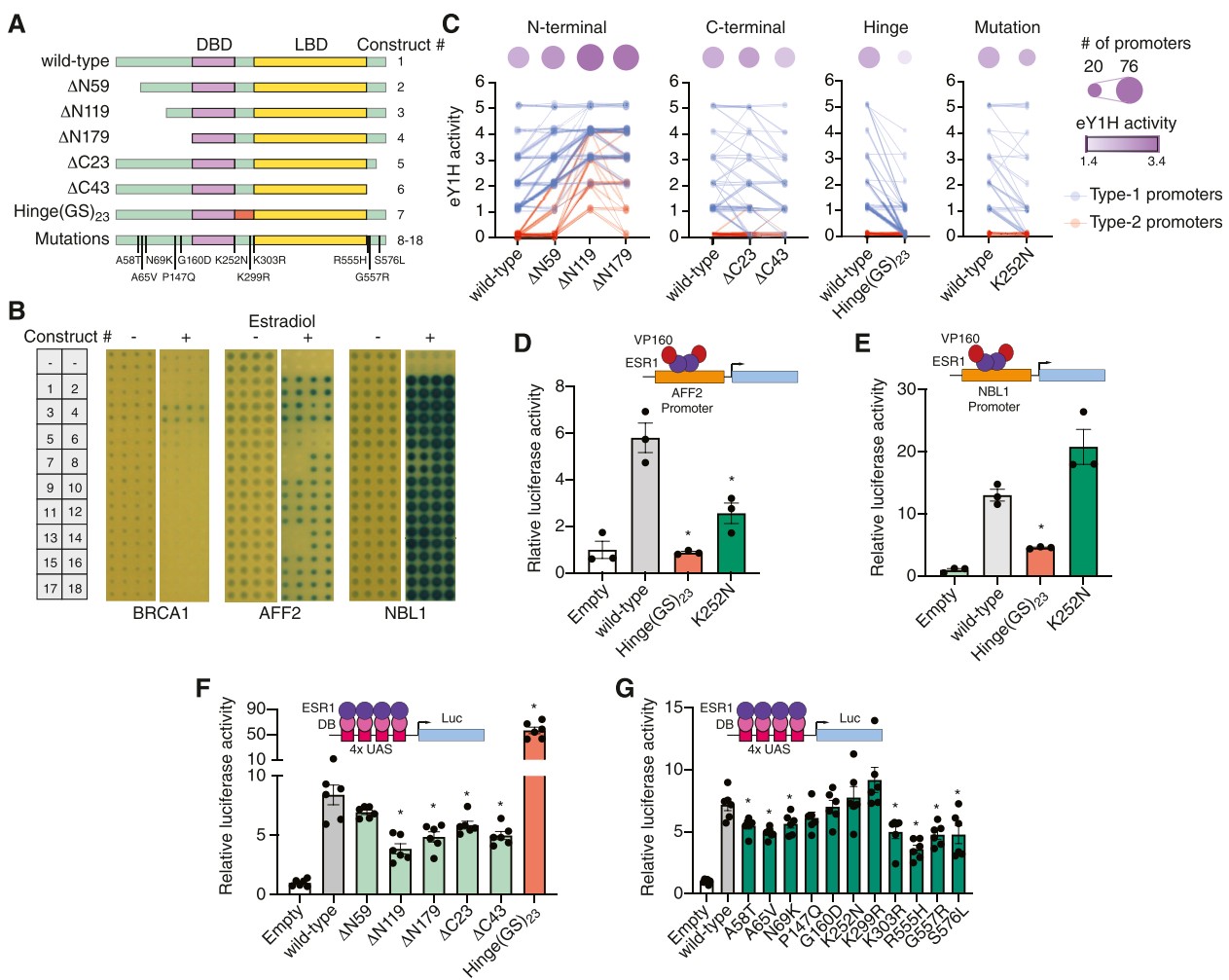

**Figure 5. Role of ESR1 intrinsically disordered regions in DNA binding and transcriptional activity.**
**(A)** Schematic of ESR1 constructs used. IDRs are indicated in green, DNA binding domain in purple, and ligand binding domain in yellow. **(B)** Examples of eY1H screens for binding of 18 different ESR1 constructs to the promoters of *BRCA1*, *AFF2*, and *NBL1* in the presence or absence of 100 nM estradiol. **(C)** eY1H binding activity scored from 0 (no binding) to 5 (very strong binding) for different ESR1 constructs. Connected lines correspond to the same cancer gene promoter. Circle sizes indicate the number of bound cancer promoters, whereas color intensity indicates the average eY1H activity across bound promoters. Type 1 promoters (blue) are those that wild-type ESR1 binds; type 2 promoters (red) are those that wild-type ESR1 does not bind. **(D, E)** Luciferase assays in HEK293T cells where the promoters of *AFF2* (D) or *NBL1* (E) were cloned upstream of firefly luciferase. ESR1 constructs were fused to 10 copies of the VP16 activator domain. Experiments were conducted in biological triplicates. *$P < 0.05$. Statistical significance was determined by a two-tailed $t$ test. **(F, G)** Mammalian one-hybrid assays measuring the transcriptional activity of different ESR1 constructs. ESR1 fusions with the Gal4 DNA binding domain (DB) are recruited to four copies of the Gal4 binding site (UAS) cloned upstream of firefly luciferase. Experiments were conducted in biological sextuplicates. *$P < 0.05$. Statistical significance was determined by a two-tailed $t$ test.

mutations, if functional, likely affect other ESR1 molecular functions such as interactions with other TFs and cofactors. A notable exception was the K252N mutation, located at the N-terminal boundary of the hinge region, which reduced binding to promoters with weak/moderate wild-type ESR1 binding but did not affect binding to promoters with strong ESR1 binding (Fig 5C). For example, the K252N mutation disrupted binding of ESR1 to the *AFF2* promoter in both eY1H and reporter assays (Fig 5B and D), whereas this mutation had no effect on ESR1 binding to the *NBL1* promoter (Fig 5B and E). Altogether, these results suggest that although the IDRs, in particular the N-terminal and hinge regions, have a strong effect on DNA binding, point mutations in these regions generally have no or only mild effects.

## ESR1 IDRs affect transcriptional activity

To evaluate whether the ESR1 mutations in the IDRs, and ESR1 IDRs in general, affect transcriptional activity, we performed mammalian one-hybrid assays in HEK293T cells. In these assays, cells were transfected with the different ESR1 constructs fused to the Gal4 DNA binding domain and a luciferase reporter vector driven by a minimal promoter and four copies of the Gal4 DNA binding site. Cells were then stimulated with 100 nM estradiol for 18 h followed by measurement of luciferase activity. Progressive truncations of the N-terminal and C-terminal IDRs resulted in reduced transcriptional activity, consistent with these regions harboring the AF-1 and AF-2 activation domains (Fig 5F). Interestingly, the Hinge(GS)$_{23}$ replacement led to a sevenfold increase in transcriptional activity.

These results show that there may be a trade-off, at least for the N-terminal IDR and hinge region, between the effects of ESR1 IDRs on DNA binding and transcriptional activity. Although the N-terminal IDR suppresses DNA binding, it contributes to transcriptional activation. Conversely, although the hinge region enhances DNA binding, it reduces transcriptional activation.

Next, we tested the effect of cancer-associated mutations in the IDRs on transcriptional activity using mammalian one-hybrid assays. Contrary to what we observed for DNA binding, multiple mutations (7/11) significantly reduced transactivation (Fig 5G). These mutations resided in the N-terminal (3/5) and C-terminal IDRs (3/3), as well as a mutation in the hinge region (1/3). Interestingly, the K252N mutation that affected DNA binding did not affect transcriptional activity. These results show that mutations in the IDRs mostly affected the transactivation function of ESR1, which is consistent with the direct involvement of IDRs in precise protein–protein interactions. The relative conservation of DNA binding function after the IDR point mutations tested suggests that larger changes in IDRs may be needed to affect DNA binding.

## Discussion

In this study, we have delineated a large-scale cancer network involving 1,350 TF-DNA interactions between 265 TFs and the promoters of 108 genes. About half of the interactions detected were previously identified in ChIP-seq experiments or were reported in the literature, illustrating the high quality of our network, while also identifying novel interactions. In particular, our network expands our knowledge of cancer gene regulation by identifying interactions involving TFs not previously known to regulate cancer-related genes. These newly associated TFs have a similar mutation rate in cancer to TFs known to regulate cancer genes, illustrating how our network can also nominate novel TFs involved in cancer. Furthermore, for many of these TFs, such as TFEC, IRF5, and ERF, we found a significant association between TF expression and cancer prognosis using TCGA data (Table S6), suggesting that the dysregulation of these TFs can also impact cancer outcomes.

Using the cancer TF-DNA interaction network, we identified TF hubs that bind to the promoters of multiple cancer-related genes. These hubs had a similar likelihood than non-hub TFs to be highly mutated in cancer, consistent with previous observations in protein–protein interaction networks that hubs are not enriched in disease-associated genes, but rather in essential genes (Goh et al, 2007; Barabasi et al, 2011). We also found that TF hubs generally bind to the promoters of both poor and good prognosis genes. Altogether, our findings suggest that TF hubs in the cancer network are unlikely to be suitable drug targets for cancer therapeutics both because of pleiotropy and because of an unclear effect on prognosis. It also further suggests that drugs used to target some of these TFs (e.g., agonists and antagonists of NHRs) in autoimmune and inflammatory conditions may also affect tumor cells in cancer patients receiving these treatments (Patalano et al, 2023). However, targeting TFs could be a suitable strategy when the goal is to reduce the expression of a single (or few) mutated oncogene(s). This will require that targeting the TF has limited side effects and that it does not lead to an unfavorable expression balance between genes associated with good and poor prognosis. In this study, we identified 25 oncogenes whose expression could potentially be targeted by inhibiting an activator/bifunctional TF that has a positive prognosis score, and is therefore also likely to reduce the expression of poor prognosis genes. Whether this ultimately leads to the expected changes in expression and results in reduced proliferation, reduced migration, or increased cell death remains to be determined.

Our study reveals that cooperativity and antagonism between TFs may play an extensive role in the regulation of cancer-related genes. This could limit the efficacy of therapeutics involving the activation or overexpression of individual TFs, as the activation of one TF in a cooperative pair may not be sufficient to induce promoter targeting, whereas an antagonized TF could still be prevented from binding despite activation. Alternative approaches may involve combinatorial treatments targeting both TFs in key cooperative pairs and antagonistic TFs rather than TFs that directly bind the promoters of interest.

An intriguing aspect of our study is the evaluation of the role of TF IDRs, particularly focusing on ESR1, in DNA binding and transcriptional activity. Our findings show that IDRs, even the ones that are not in close proximity to a DNA binding domain, can modulate DNA binding by both increasing or decreasing the number and strength of interactions with target gene promoters, consistent with previous studies (Brodsky et al, 2021). Our ESR1 results are also consistent with a more general pattern showing that isoforms of different TFs, such as MAX, STAT1, and RXRG, with intact DNA binding domains can have different DNA targets and different functional relationships with interacting TF partners (Berenson et al, 2023). Interestingly, IDRs that do not overlap with mapped effector domains can also impact transcriptional activity. In particular, replacing the hinge region of ESR1 for a flexible $(GS)_{23}$ linker increased activation by sevenfold while significantly reducing DNA binding. This suggests not only that IDRs modulate TF functions, but also that some IDRs may contribute to trade-offs between DNA binding and activity.

The use of yeast-based systems for analyzing TF-DNA interactions, while powerful, may not fully recapitulate the contexts in which they occur in cancer cells, but rather provides a repertoire of possible interactions for future study. In addition, although our resource of 700 promoter clones is of great utility to the community studying cancer gene regulation, other distal elements such as enhancers and silencers may significantly contribute to the regulation of these cancer genes. Future studies could expand our resources and extend these analyses to encompass a wider range of regulatory elements. Overall, our work provides an experimental and informational resource that can facilitate and motivate future investigations of the role of TFs in cancer gene dysregulation. Such studies will open the door to explore the targeting of TFs as an avenue for cancer treatment.

## Materials and Methods

### Generation of entry clones and integrant yeast strains for cancer-related gene promoters

Entry clones and yeast strains for promoters of cancer-related genes were generated following the established procedures

(Fuxman Bass et al, 2016a, 2016b). We selected 697 genes from the Cancer Gene Census (Tate et al, 2019), as well as 114 additional genes whose expression is often dysregulated in cancer. For 190 of these genes, we included alternative promoters in cases where H3K27ac or H3K4me3 marks were observed in data from the ENCODE Project (ENCODE Project Consortium, 2012). Promoters encompassing approximately 2 kb upstream of the transcription start site were amplified from human genomic DNA (Clontech) using primers flanked with Gateway tails (Tables S1 and S7). After two rounds of cloning attempts, we successfully cloned 700 promoter sequences corresponding to 556 cancer-related genes into the pDONR-P4P1R vector using BP Clonase (#11789100; Thermo Fisher Scientific), resulting in a collection of Gateway entry clones whose sequences were verified through Sanger sequencing (Sequegen). Subsequently, each promoter was transferred to the pMW#2 (#13349; Addgene) and pMW#3 (#13350; Addgene) destination vectors using LR Clonase (#11791100; Thermo Fisher Scientific), positioning them upstream of the *HIS3* and *LacZ* reporter genes, respectively. Destination vectors were linearized using single-cutter restriction enzymes (R0520L, R0146L, R3127S, R0581S, R0193L, R0114S, R0187S, R0519L; New England Biolabs).

The pWM#2 and pWM#3 plasmids for each cancer promoter were integrated simultaneously into the Y1Has2 yeast genome, as previously outlined (Reece-Hoyes et al, 2011b; Fuxman Bass et al, 2016b) and as described below. Yeast were cultured in 1 liter liquid YAPD media at 30°C with shaking at 200 rpm until reaching OD600 = 0.5, followed by washing with sterile water and 1X TE + 0.1 M lithium acetate (TE/LiAc). Yeast were resuspended in TE/LiAc with salmon sperm DNA (15632011; Thermo Fisher Scientific) at a dilution of 1:10, and 2 µg of each digested plasmid (pWM#2 and pWM#3) was added. Six volumes of TE/LiAc + 40% polyethylene glycol were added and gently mixed 10 times, followed by a first incubation at 30°C for 30 min and a second incubation at 42°C for 20 min. The yeast were then resuspended in sterile water and plated on selective media lacking histidine and uracil to select for double integrants.

### Sequence confirmation of cancer gene promoter yeast strains

Cancer gene promoter yeast strains were sequence-confirmed using the SWIM-seq protocol (Luck et al, 2020). In brief, yeast were treated with zymolyase (0.2 KU/ml) (Z1004; United States Biological) for 30 min at 37°C followed by 10 min at 95°C to disrupt cell walls and release DNA. Promoter sequences were PCR-amplified in a 96-well format using forward primers with well-specific barcodes. The primer design is shown as follows:

Forward primer (pMW#2):
5′-AGACGTGTGCTCTTCCGATCT[barcode]GGCCGCCGACTAGTGATA-3′
Reverse primer (pMW#2):
5′-GGGACCACCCTTTAAAGAGA-3′
Forward primer (pMW#3):
5′-AGACGTGTGCTCTTCCGATCT[barcode]GCCAGTGTGCTGGAATTCG-3′
Reverse primer (pMW#3):
5′-ATCTGCCAGTTTGAGGGGAC-3′

PCRs were conducted using DreamTaq Polymerase (EP0705; Thermo Fisher Scientific) under the following conditions: 95°C for 3 min; 35 cycles of 95°C for 30 s, 56°C for 30 s, and 72°C for 4 min; and final extension at 72°C for 7 min. Amplicons from each 96-well plate were pooled and purified using PCR Purification Kit (K310002; Thermo Fisher Scientific). Each pooled sample was prepared as a single sequencing library by the Molecular Biology Core Facilities at the Dana-Farber Cancer Institute; DNA was sheared using an ultrasonicator (Covaris) before tagmentation. Libraries were sequenced using NovaSeq with ~10 million reads (paired-end, 150 bp) per library. For a promoter yeast strain to be confirmed, we required at least 25% of sequencing reads for the pMW#3 vector to align with the expected promoter sequence. Sequencing data can be found at the NCBI Sequence Read Archive at the accession number PRJNA1015222.

### eY1H screening

We performed eY1H assays using a human TF yeast array (Reece-Hoyes et al, 2011a) as previously described and as follows using a high-density array ROTOR robot (Singer Instruments). The three-plate human TF yeast array and promoter yeast strains were mated pairwise on permissive medium agar plates and incubated at 30°C for 1 d. Mated yeast were then transferred to selective medium agar plates lacking uracil and tryptophan to select for successfully mated yeast and incubated at 30°C for 2 d. Diploid yeast were finally transferred to selective medium agar plates lacking uracil, tryptophan, and histidine, with 5 mM 3AT and 320 mg/liter X-gal. Readout plates were imaged 2, 3, 4, and 7 d after final plating. Each interaction was tested in adjacent quadruplicate colonies. Interactions were considered positive if at least three colonies displayed reporter activity. Results are reported in Table S2.

### pY1H screening

We performed pY1H assays using a previously generated TF-pair array (Berenson et al, 2023). Screening of TF pairs and cancer gene promoters was performed similar to eY1H screens as previously described and as follows using a high-density array ROTOR robot (Singer Instruments). The five-plate TF-pair yeast array and promoter yeast strains were mated pairwise on permissive medium agar plates and incubated at 30°C for 1 d. Mated yeast were then transferred to selective medium agar plates lacking uracil, leucine, and tryptophan to select for successfully mated yeast and incubated at 30°C for 2 d. These selection plates were imaged and analyzed to identify array locations with failed yeast growth, which were then removed from further analysis. Diploid yeast were finally transferred to selective medium agar plates lacking uracil, leucine, tryptophan, and histidine, with 5 mM 3AT and 320 mg/liter X-gal. Readout plates were imaged 2, 3, 4, and 7 d after final plating. Each interaction was tested in adjacent quadruplicate colonies. For each TF-pair strain, corresponding TF1 and TF2 single-TF strains are included in the same plate. Results are reported in Tables S2 and S3.

### Identifying cooperative and antagonistic interactions

Yeast plate images were processed and visualized using DISHA (Detection of Interactions Software for High-throughput Analyses) software as previously described (Berenson et al, 2023). TF-pair strains were sorted based on each index (cooperativity, antagonism index 1, and antagonism index 2) separately. Images were then

manually analyzed to identify cooperative and antagonistic interactions. To call an interaction, we required the following criteria:

1. TF-pair, TF1, and TF2 yeast strains all showed growth in the mating selection plates before transfer to readout plates.

2. On readout plates, ≥3 out of 4 quadruplicate colonies were uniform for TF-pair, TF1, and TF2 yeast strains.

3. For cooperative interactions, TF-pair yeast showed a strong or moderate reporter activity relative to the empty–empty strain. TF1 and TF2 yeast showed no or only weak reporter activity.

4. For antagonistic interactions, TF1 and/or TF2 yeast showed a strong or moderate reporter activity relative to the empty–empty strain. TF-pair yeast showed no or only weak reporter activity.

### Literature and ChIP-seq evidence for interactions detected by eY1H and pY1H assays

Literature evidence for eY1H- and pY1H-derived interactions was determined by performing searches in the PubMed database. If there was at least one piece of experimental evidence indicating the binding or regulation of the TF to the cancer promoter or regulation of the cancer gene, then the TF–gene interaction was considered to be previously reported. Results are reported in Table S2.

ChIP-seq data were downloaded from the GTRD (Yevshin et al, 2019) in MACS2 (Zhang et al, 2008) peak calling result format. If a peak was called in ChIP-seq data for a given TF and the center of the peak was within the corresponding promoter region, the TF was considered to bind the promoter. Results are reported in Table S2. The code for this analysis is available in Lu et al (2024).

### TF and cancer gene survival analysis

RNA-seq data associated with clinical data from 33 tumor types were downloaded from TCGA and organized using TCGAbiolinks (Colaprico et al, 2016). Expression data were then normalized using the CPM method and $\log_2$-transformed. To determine whether the expression levels of cancer-related genes and TFs in our eY1H-derived network were associated with good or poor prognosis, survival analyses were conducted using the normalized RNA-seq data for each tumor type. First, the Cox proportional-hazards model was used to test whether the high/low expression level of the gene or TF will impact survival significantly (adjusted $P < 0.05$); the hazard ratio from the Cox proportional-hazards model indicates whether the high or low expression of the gene leads to good or poor prognosis. An ANOVA was then used to regress out the confounding factors of age, gender, race, tumor size, tumor metastasis, and tumor stage. All the survival analyses were performed using the survival package (Therneau et al, 2000). Results are reported in Table S6. The code for this analysis is available in Lu et al (2024).

### Determining mutation incidence of TFs

The COSMIC database (Sondka et al, 2018) was used to determine the number of cancer cases in which mutations have been observed for each TF. The mutation frequency was calculated as the total number of cases with mutations minus the number of synonymous mutations, divided by the total number of all cases. Information can be found in Table S8.

### Permutation analysis

The prognosis labels of the genes were randomly shuffled 1,000 times, and then, the prognostic score for each TF was calculated in the randomized networks. The values for TFs of each degree in the real network were compared with those in the randomized network for the same degree.

### Cancer expression analysis

The transcriptome datasets of primary tumors were obtained from TCGA using TCGAbiolinks package in R (Colaprico et al, 2016). To determine the expression levels of each TF in each cancer type, the transcriptome dataset was normalized using the CPM method (Robinson et al, 2010). Then, the median CPM for each TF was determined across samples belonging to a cancer type. Information can be found in Table S9.

To determine differential TF expression in primary tumors, DESeq2 (Love et al, 2014) was used for differential expression analysis between primary tumor samples and normal tissue samples from TCGA datasets. Analyses were conducted only if there were more than 10 normal tissue samples available. To minimize false positives because of sparse data, TFs were tested only if they had more than 1 CPM in more than 10% of the primary tumor samples. Information can be found in Table S10.

### ESR1 DNA constructs

The COSMIC database (Sondka et al, 2018) was used to identify mutations in the ESR1 IDRs occurring in breast cancer patients and across cancers. Mutations occurring in at least two patients were selected to be tested by eY1H and mammalian one-hybrid (M1H) assays. Deletion constructs were selected to cover portions of the N- and C-terminal IDRs to better identify regions that affect DNA binding and transcriptional activity. The hinge region was replaced by a flexible linker consisting of 23 glycine–serine repeats to maintain the length and flexibility of this region. All ESR1 constructs were ordered from GenScript in pUC57 vectors. The ESR1 sequences were flanked by attB1 and attB2 sequences for Gateway cloning into the pDONR221 entry vector, as well as destination vectors for eY1H and M1H assays. Information and sequences for ESR1 constructs can be found in Tables S11 and S12.

### Generation of DB-pEZY3 and 4xUAS-pGL4.23 vectors for M1H assays

The DB-pEZY3 and 4xUAS-pGL4.23 vectors were generated for M1H assays. To generate the DB-pEZY3 vector, the coding sequence for the yeast Gal4 DNA binding domain (DBD) was cloned into the pEZY3 mammalian expression vector upstream of the insert region. Proteins cloned into the insert region are therefore expressed with the Gal4 DBD fused to the N-terminus. To generate the 4xUAS-pGL4.23 vector, four copies of the yeast upstream activating sequence (UAS) site were cloned into the pGL4.23 vector upstream of the minimal promoter and firefly luciferase reporter gene. The UAS

site is recognized by the Gal4 DBD and therefore recruits any protein expressed as a fusion with the Gal4 DBD.

## Cloning of ESR1 constructs

ESR1 constructs were cloned into the pDONR221 entry vector using BP Clonase (#11789100; Thermo Fisher Scientific) and verified by whole plasmid sequencing (Plasmidsaurus) and Sanger sequencing (Genewiz) to confirm the proper mutant insertion and discard clones with additional unwanted mutations. Confirmed entry vectors were cloned into the pAD2$\mu$ (Walhout Lab), pEZY3-VP160, and DB-pEZY3 destination vectors using LR Clonase (#11791100; Thermo Fisher Scientific). Plasmid samples were prepared using Endotoxin-Free Miniprep Kit (#W210650; 101 BIO) following the supplier's protocol.

## Generation of yeast strains expressing ESR1 constructs

Each pAD2$\mu$ vector carrying a cloned ESR1 construct was transformed into Y$\alpha$1867 yeast as previously outlined (Reece-Hoyes et al, 2011a; Fuxman Bass et al, 2016b) and as described below. Yeast were cultured in 1 liter liquid YAPD media at 30°C with shaking at 200 rpm until reaching OD600 = 0.5, followed by washing with sterile water and 1X TE + 0.1 M lithium acetate (TE/LiAc). Yeast were resuspended in TE/LiAc with salmon sperm DNA (15632011; Thermo Fisher Scientific) at a dilution of 1:10, and ~250 ng of pAD2$\mu$ plasmid was added. Six volumes of TE/LiAc + 40% polyethylene glycol were added and gently mixed 10 times, followed by a first incubation at 30°C for 30 min and a second incubation at 42°C for 20 min. The yeast were then resuspended in sterile water and plated on selective media lacking tryptophan to select for transformants.

## eY1H screening of ESR1 constructs

We performed eY1H assays as follows using a high-density array ROTOR robot (Singer Instruments). Y$\alpha$1867 yeast strains transformed with ESR1 construct clones were arrayed such that all 18 ESR1 constructs and two empty control yeast strains were tested against 10 different promoters in each 1,536-colony agar plate. ESR1 construct yeast strains and all 508 cancer gene promoter yeast strains were mated pairwise on permissive medium agar plates and incubated at 30°C for 1 d. Mated yeast were then transferred to selective medium agar plates lacking uracil and tryptophan to select for successfully mated yeast and incubated at 30°C for 2 d. Diploid yeast were finally transferred to selective medium agar plates lacking uracil, tryptophan, and histidine, with 5 mM 3AT and 320 mg/liter X-gal. Readout plates were imaged 2, 3, 4, and 7 d after final plating. Each interaction was tested in adjacent quadruplicate colonies and scored manually on a scale from 0 (no reporter signal) to 5 (very strong binding signal). Results are reported in Table S5.

## Confirmation of eY1H interactions using mammalian luciferase assays

Cancer gene promoter sequences were cloned from the pDONR-P4P1R entry vector into the pGL4.23 vector upstream of a minimal promoter and luciferase reporter gene using LR Clonase (#11791100; Thermo Fisher Scientific). ESR1 constructs were cloned from the pDONR221 entry vector into the pEZY3-VP160 expression vector using LR Clonase (#11791100; Thermo Fisher Scientific) to express each ESR1 construct as a fusion with 10 copies of the VP16 activation domain.

Luciferase assays were conducted in HEK293T cells (#CRL-11268; ATCC) to identify interactions between each ESR1 construct and a subset of cancer gene promoters. Cells were cultured in DMEM (#11965118; Gibco) with 10% fetal bovine serum (#S12450H; Bio-Techne) and 1% antibiotic–antimycotic (#15240062; Gibco) at 37°C with 5% CO2. Cells were plated at a density of ~10,000 cells/well in 96-well white opaque sterile plates (#25382-208; Falcon) with growth media and incubated for 24 h. Cells were transfected with Lipofectamine 3000 (#L3000001; Invitrogen) following the manufacturer's protocol using an 80 ng pEZY3-VP160 vector with a cloned ESR1 construct, a 60 ng pGL4.23 vector with a cloned cancer gene promoter, and a 10 ng Renilla-pGL4.74 vector. Three biological replicates were performed for each construct, and an empty pEZY3-VP160 vector with no cloned ESR1 construct was used as a negative control. Cells were incubated for 6 h, treated with 100 nM estradiol, and incubated for an additional 18 h.

Luciferase assays were performed using Dual-Glo Luciferase Assay System (#E2940; Promega) following the manufacturer's protocol. Luminescence was measured on a Victor3 multilabel reader (#1420; PerkinElmer) using Renilla and firefly filters. Background signal from untransfected cells was subtracted from each Renilla and firefly measurement. Firefly/Renilla ratios for each sample were normalized to the average ratio for negative control samples transfected with the empty pEZY3-VP160 vector.

## Mammalian one-hybrid (M1H) assays of ESR1 constructs

M1H assays were conducted in HEK293T cells (#CRL-11268; ATCC) to identify transcription-activating or transcription-repressing functions of our ESR1 constructs. Cells were cultured in DMEM (#11965118; Gibco) with 10% fetal bovine serum (#S12450H; Bio-Techne) and 1% antibiotic–antimycotic (#15240062; Gibco) at 37°C with 5% CO$_2$. Cells were plated at a density of ~10,000 cells/well in 96-well white opaque sterile plates (#25382-208; Falcon) with growth media and incubated for 24 h. Cells were transfected with Lipofectamine 3000 (#L3000001; Invitrogen) following the manufacturer's protocol using an 80 ng DB-pEZY3 vector with a cloned ESR1 construct, a 60 ng 4xUAS-pGL4.23 vector, and a 10 ng Renilla-pGL4.74 vector. Three biological replicates were performed for each construct, and an empty DB-pEZY3 vector with no cloned ESR1 construct was used as a negative control. Cells were incubated for 6 h, treated with 100 nM estradiol, and incubated for an additional 18 h.

Luciferase assays were performed using Dual-Glo Luciferase Assay System (#E2940; Promega) following the manufacturer's protocol. Luminescence was measured on a Victor3 multilabel reader (#1420; PerkinElmer) using Renilla and firefly filters. Background signal from untransfected cells was subtracted from each Renilla and firefly measurement. Firefly/Renilla ratios for each sample were normalized to the average ratio for negative control samples transfected with the empty DB-pEZY3 vector.

# Data Availability

All data generated in this study are available within the article and its supplementary information.

# Supplementary Information

# Acknowledgements

We thank Drs. Martha Bulyk and Haribabu Arthanari, and members of the Fuxman Bass laboratory for helpful discussions regarding experiments and data analysis. This work was funded by the National Institutes of Health grants R35 GM128625 and U01 CA232161, awarded to JI Fuxman Bass; and R35 GM147254, awarded to A Fiszbein.

## Author Contributions

Y Lu: conceptualization, data curation, formal analysis, investigation, visualization, methodology, and writing—original draft, review, and editing.

A Berenson: conceptualization, data curation, formal analysis, investigation, methodology, and writing—original draft, review, and editing.

R Lane: data curation, formal analysis, investigation, methodology, and writing—review and editing.

I Guelin: investigation, methodology, and writing—review and editing.

Z Li: formal analysis, investigation, visualization, methodology, and writing—review and editing.

Y Chen: investigation, methodology, and writing—review and editing.

S Shah: investigation, methodology, and writing—review and editing.

M Yin: investigation, methodology, and writing—review and editing.

LF Soto-Ugaldi: formal analysis, investigation, methodology, and writing—review and editing.

A Fiszbein: resources, supervision, funding acquisition, and writing—review and editing.

JI Fuxman Bass: conceptualization, resources, data curation, formal analysis, supervision, funding acquisition, investigation, visualization, methodology, project administration, and writing—original draft, review, and editing.

## Conflict of Interest Statement

The authors declare that they have no conflict of interest.

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
