## [Reviewer comments · Life Science Alliance]

Life Science Alliance

A large-scale cancer-specific protein-DNA interaction network

Yunwei Lu, Anna Berenson, Ryan Lane, Isabelle Guelin, Zhaorong Li, Yilin Chen, Sakshi Shah, Meimei Yin, Luis Fernando Soto-Ugaldi, Ana Fiszbein, and Juan Fuxman Bass

DOI: <https://doi.org/10.26508/lsa.202402641>

Corresponding author(s): Juan Fuxman Bass, Boston University

Review Timeline:

Submission Date:	2024-02-03
Editorial Decision:	2024-04-12
Revision Received:	2024-05-28
Editorial Decision:	2024-07-02
Revision Received:	2024-07-03
Accepted:	2024-07-04

Transaction Report:

April 12, 2024

Re: Life Science Alliance manuscript #LSA-2024-02641-T

Assistant Professor Juan Fuxman Bass
Boston University

Dear Dr. Fuxman Bass,

Thank you for submitting your manuscript entitled "A large-scale cancer-specific protein-DNA interaction network" to Life Science Alliance. The manuscript was assessed by expert reviewers, whose comments are appended to this letter. We invite you to submit a revised manuscript addressing the Reviewer comments.

Thank you for this interesting contribution to Life Science Alliance. We are looking forward to receiving your revised manuscript.

Sincerely,

B. MANUSCRIPT ORGANIZATION AND FORMATTING:

Reviewer #1 (Comments to the Authors (Required)):

This manuscript by Lu et al. demonstrates a large-scale cancer gene transcription factor-DNA interaction network and investigates the role of IDR in DNA binding and transcriptional activity. This study is important and provide data resource of the cancer gene regulation for the scientific community

(1) In Figure 1, although the author classifies cancers based on their origin, it would be very helpful to classify them based on drug sensitivity. For example, FDA-approved PARP inhibitor sensitivity is an important criterion for studying the number of genes associated with this category.

(2) Genes involved in tandem duplication and translocation also play a role in cancer progression. In Figure 1, Panel B- authors can include those in study.

(3) The Enhancer impacts gene regulation and cancer, and the same is true for the CTCF boundary element. Please comment on it.

(4) It would be helpful if the authors showed not only the binding of Transcriptional factors but also their upregulation and downregulation.

(5) BRCA1, a well-known tumor suppressor gene, also has intrinsically disordered regions. Please include that in study.

Reviewer #2 (Comments to the Authors (Required)):

The authors generated a resource of cancer gene promoter reporters for yeast one-hybrid assays and apply it to two experiments: a large-scale screen of TFs and, to a lesser extent, TF pairs; and a more focused structure-function study of the effect on DNA binding and transcriptional activation by various alterations of the disordered regions of ESR1 protein. The results of the first screen suggest some TFs that could represent targets for interventions to improve patient prognosis, but also cast significant doubt on the utility of this approach overall. The part of the manuscript that presents on the effects of deletions versus cancer mutations in the intrinsically disordered regions (IDRs) of ESR1 on DNA binding versus transcriptional activity is particularly interesting.

Overall, this study provides a valuable resource for various aspects of cancer genetics and generates many important hypotheses for follow-up, and we are supportive of this manuscript being eventually published in LSA based on the experimental findings. However, in its current state it is marred by significant "overselling" and requires rewriting to tone down the language, plus it requires some statistical analysis, as described below.

The first main issue is that it is not clear that a "network" paradigm is the most appropriate way to present these data. Most of the experiments reported here are binary, and the question of which assayed TFs are ever co-expressed in a cancer cell context is not explored, so it's not clear how network effects of the TF-promoter interactions or concepts like "degree" are relevant to this system.

The second is that many parts of the manuscript seem to be written "backwards," with significant caveats or explanations provided only long after a point is introduced. Examples include:

- The observation on lines 207-210 that TFs with fewer targets have more instances where targets belong to predominantly one prognosis class. This seems like what you would expect to see by chance, but the possibility that sampling more targets would reveal a more balanced distribution is not raised until the discussion.

- The "network" is based entirely on promoter binding, but the existence of distal regulatory elements is not mentioned until the final paragraph of the discussion.

Third, there are mentions throughout the results of "overrepresentation" or "depletion" (e.g., in lines 185, 195, 208, 225, 302) with no comparison to what would be expected by chance in a dataset of this size. The statistical significance of potential overrepresentation or deletion should be calculated and reported along with the statistical test that is used.

More specific points to address:

1. It would be good to provide more description of the library design in the first Results section, "Generation of a comprehensive clone resource of cancer gene promoters". Do the 700 promoters of 556 cancer-related genes represent all TSSs for those

genes, and if not, then how were they chosen? A description of what constitutes a "cancer-related gene" for the purposes of this study can be inferred from some of the following paragraph, but it would be better if it were stated explicitly.

2. Discussion of panels D and E of Figure 1 inverts them.

3. The use of several pale, low-saturation colors for the edges in Figure 2A makes them indistinguishable to a reader with even partial colorblindness.

4. The "111 interactions that were reported in the literature" listed in Supplementary Table 2 should include references (PMIDs would be sufficient).

5. There is surprisingly little discussion of the "631 novel interactions" observed. How many of these represent TFs for which high-quality target identification data were not previously available, versus how many correspond to binding events of TFs that have previously been assayed by genome-wide location methods and not observed to bind there? Is there any other evidence to support the in vivo relevance of these interactions?

6. The section on cooperative vs. antagonistic binding is notably light on references. Have any of these TF pairs besides MYC-MAX been explored for non-additive interactions before?

7. The statement in line 383 that "these observations were confirmed" by luciferase assay is overly broad, since it appears to apply to all three disordered regions of ESR1. The preceding paragraph had stated that increasingly large deletions of the N-terminal IDR led to increased DNA binding, but the two promoters shown in Figure 5D,E show significantly decreased and unchanged binding, respectively.

8. It would be helpful to mention in the Methods how many replicates were performed for each test and if they were all performed simultaneously on one plate. Similarly, it seems that the phenotypic classes in Figure 5C are scored algorithmically by the DISHA software; if so, it would be helpful to state this explicitly when they're introduced.

9. The Supplementary Data statement should be updated.

Response to the reviewers

Reviewer #1 (Comments to the Authors (Required)):

This manuscript by Lu et al. demonstrates a large-scale cancer gene transcription factor-DNA interaction network and investigates the role of IDR in DNA binding and transcriptional activity. This study is important and provide data resource of the cancer gene regulation for the scientific community

We thank the reviewer for the supportive comments on the manuscript.

(1) In Figure 1, although the author classifies cancers based on their origin, it would be very helpful to classify them based on drug sensitivity. For example, FDA-approved PARP inhibitor sensitivity is an important criterion for studying the number of genes associated with this category.

We appreciate the suggestion and agree that it can be useful to classify cancers based on drug sensitivity. However, this is beyond the scope of this study, which provides a broad resource for the study of cancer genes across many cancer types.

(2) Genes involved in tandem duplication and translocation also play a role in cancer progression. In Figure 1, Panel B- authors can include those in study.

The reviewer has made a great suggestion. We included the information about translocation in our updated figure 1B. As tandem duplication is not our focus in this study (and also not reported in CGC), we did not include it in this figure.

(3) The Enhancer impacts gene regulation and cancer, and the same is true for the CTCF boundary element. Please comment on it.

This is an important point by the reviewer. We have discussed the role of distal elements in cancer gene regulation at the beginning of the results section of the revised manuscript, explaining our decision to focus on promoter regions rather than other regulatory elements as follows (lines 77-82):

“Systematic studies of TF-DNA binding and transcriptional activity often require large-scale clone resources of regulatory DNA elements - such as promoters and enhancers - that can be tested across functional assays. Considering that promoters are the primary drivers of gene expression while enhancers require more specific spatiotemporal contexts to function effectively, we focused on promoter regions of cancer-related genes to study their regulation by TFs (Bergman et al. 2022).”

(4) It would be helpful if the authors showed not only the binding of Transcriptional factors but also their upregulation and downregulation.

We thank the reviewer for this suggestion. To address this, first, we determined the expression across cancer types in TCGA for TFs absent from the network and TFs with different degrees. We found that highly connected TFs generally had higher expression than those lowly connected or absent from the network. We also performed differential expression analysis for each TF between tumor and normal control samples from TCGA. We observed that multiple TFs in our network were dysregulated in each cancer type. This suggests that, in addition to being expressed, dysregulation of these TFs may propagate in the network and affect the expression of cancer-related target genes. These results are depicted in Figure 2H and Supplementary Figure 2 and is discussed in the Results section (lines 180-191):

“TFs in the cancer network are generally more highly expressed across cancers than TFs for which we did not detect any interactions (Figure 2H). Further, TFs in the network also tend to be more often differentially expressed in tumor samples versus matched normal controls (Supplementary Figure 2). Together, this suggests that TFs in the cancer TF-DNA network are cancer-relevant.

TFs in the network bind to a widely different number of promoters, ranging from 1 to 54 promoters. Half (117/265) of TFs bind to just one promoter, while 13.58% (36/265) bind to 10 or more (Figure 2A). This is consistent with a power-law distribution, which is frequently observed in gene regulatory networks (Supplementary figure 1D) (Lima-Mendez and van Helden 2009). TFs absent from the network and those with 1 or few interactions have overall lower expression levels across cancer types than highly connected TFs (TF hubs) (Figure 2H). This suggests that TF hubs (i.e., highly connected TFs) may have a substantial impact on gene regulatory networks and represent potentially valuable drug targets as they coordinate the expression of multiple genes.”

(5) BRCA1, a well-known tumor suppressor gene, also has intrinsically disordered regions. Please include that in study.

As the reviewer commented, many proteins have intrinsically disordered regions (IDRs). Indeed, IDRs are suspected to play major roles in the function of TFs as well as many other genes. Here, we present a case study in which we identify how IDRs contribute to the function of the cancer-relevant transcription factor ESR1. Future studies will reveal the function of IDRs in other TFs and non-TF proteins such as BRCA1. In particular, we cannot evaluate BRCA1 as it does not bind to DNA directly, and hence cannot be tested by eY1H or pY1H assays.

Reviewer #2 (Comments to the Authors (Required)):

The authors generated a resource of cancer gene promoter reporters for yeast one-hybrid assays and apply it to two experiments: a large-scale screen of TFs and, to a lesser extent, TF pairs; and a more focused structure-function study of the effect on DNA binding and

transcriptional activation by various alterations of the disordered regions of ESR1 protein. The results of the first screen suggest some TFs that could represent targets for interventions to improve patient prognosis, but also cast significant doubt on the utility of this approach overall. The part of the manuscript that presents on the effects of deletions versus cancer mutations in the intrinsically disordered regions (IDRs) of ESR1 on DNA binding versus transcriptional activity is particularly interesting.

Overall, this study provides a valuable resource for various aspects of cancer genetics and generates many important hypotheses for follow-up, and we are supportive of this manuscript being eventually published in LSA based on the experimental findings. However, in its current state it is marred by significant "overselling" and requires rewriting to tone down the language, plus it requires some statistical analysis, as described below.

We thank the reviewer for the encouraging comments on the manuscript. We have addressed the main reviewer's concerns and suggestions below which we believe have further strengthened the manuscript.

The first main issue is that it is not clear that a "network" paradigm is the most appropriate way to present these data. Most of the experiments reported here are binary, and the question of which assayed TFs are ever co-expressed in a cancer cell context is not explored, so it's not clear how network effects of the TF-promoter interactions or concepts like "degree" are relevant to this system.

We thank the reviewer for pointing this out and for the excellent suggestion of looking at TF expression across cancers, which has led to new insights included in the revised manuscript. To explore the relation between the degree and TF expression, we calculated the CPM for each TF in each cancer type using samples from TCGA. We found that most TFs in our network are indeed expressed in different cancers, presenting the opportunity for these TFs to regulate the putative target genes. We also observed that the expression of TFs with interactions in the network is generally higher than that of TFs tested in the assay but without interactions. Further, TFs with high degree usually have higher expression than those with low degree. This suggests that highly connected TFs may be more central or crucial in the regulatory network as they need higher expression levels to maintain or regulate essential biological processes in cancer cells, and their dysregulation could have significant impacts on cancer progression. We further found that many of these highly connected TFs are dysregulated in cancer (compared to normal tissue controls). These results are depicted in Figure 2H and Supplementary Figure 2 and is discussed in the Results section (lines 180-191):

"TFs in the cancer network are generally more highly expressed across cancers than TFs for which we did not detect any interactions (Figure 2H). Further, TFs in the network also tend to be more often differentially expressed in tumor samples versus matched normal controls (Supplementary Figure 2). Together, this suggests that TFs in the cancer TF-DNA network are cancer-relevant.

TFs in the network bind to a widely different number of promoters, ranging from 1 to 54 promoters. Half (117/265) of TFs bind to just one promoter, while 13.58% (36/265) bind to 10 or

more (Figure 2A). This is consistent with a power-law distribution, which is frequently observed in gene regulatory networks (Supplementary figure 1D) (Lima-Mendez and van Helden 2009). TFs absent from the network and those with 1 or few interactions have overall lower expression levels across cancer types than highly connected TFs (TF hubs) (Figure 2H). This suggests that TF hubs (i.e., highly connected TFs) may have a substantial impact on gene regulatory networks and represent potentially valuable drug targets as they coordinate the expression of multiple genes."

The second is that many parts of the manuscript seem to be written "backwards," with significant caveats or explanations provided only long after a point is introduced. Examples include:

- The observation on lines 207-210 that TFs with fewer targets have more instances where targets belong to predominantly one prognosis class. This seems like what you would expect to see by chance, but the possibility that sampling more targets would reveal a more balanced distribution is not raised until the discussion.

We thank the reviewer for catching this oversight! We performed a permutation analysis in which we randomized the prognosis labels of the genes 1000 times and then calculated the prognostic score for each TF again. We noticed there wasn't any meaningful enrichment (Supplementary Figure 3), so we decided to remove this section from the revised manuscript. We have then limited our discussion to hub TFs as our previous statement that they bind to the promoters of both good and poor prognosis genes stands, which could be challenging to target with predictable outcomes. The revised results section is the following (lines 192-194):

"However, we found that none of the TFs with 10 or more interactions in our network display any significant bias toward binding to the promoters of genes associated with either good or poor prognosis (Figure 2G and Supplementary Figure 3)."

- The "network" is based entirely on promoter binding, but the existence of distal regulatory elements is not mentioned until the final paragraph of the discussion.

We have discussed the role of distal elements in cancer gene regulation at the beginning of the results section in the revised manuscript, explaining our decision to focus on promoter regions rather than other regulatory elements as follows (lines 77-82):

"Systematic studies of TF-DNA binding and transcriptional activity often require large-scale clone resources of regulatory DNA elements - such as promoters and enhancers - that can be tested across functional assays. Considering that promoters are the primary drivers of gene expression while enhancers require more specific spatiotemporal contexts to function effectively, we focused on promoter regions of cancer-related genes to study their regulation by TFs (Bergman et al. 2022)."

Third, there are mentions throughout the results of "overrepresentation" or "depletion" (e.g., in lines 185, 195, 208, 225, 302) with no comparison to what would be expected by chance in a

dataset of this size. The statistical significance of potential overrepresentation or deletion should be calculated and reported along with the statistical test that is used.

We thank the reviewer for the suggestion. For the overrepresentation or depletion of PDIs in each TF family, we performed proportion comparison tests to determine the statistical significance. For the cooperation and antagonistic events involved in each TF family, we did Fisher's exact test to determine the statistical significance. We have updated Figures 2F and 4D in the revised manuscript as well as the figure legends.

More specific points to address:

1. It would be good to provide more description of the library design in the first Results section, "Generation of a comprehensive clone resource of cancer gene promoters". Do the 700 promoters of 556 cancer-related genes represent all TSSs for those genes, and if not, then how were they chosen? A description of what constitutes a "cancer-related gene" for the purposes of this study can be inferred from some of the following paragraph, but it would be better if it were stated explicitly.

We thank the reviewer for the suggestion. We initially selected 697 genes from the Cancer Gene Census, as well as 114 additional genes whose expression is often dysregulated in cancer. In addition, we included for 190 of these genes alternative promoters, in cases where H3K27ac or H3K4me3 marks were observed in data from the Encode Project. After 2 rounds of cloning attempts, we successfully cloned 700 promoter sequences - each comprising 2 kb of sequence immediately upstream of a transcription start site - corresponding to 556 cancer-related genes. These were cloned into Gateway vectors for easy transfer into different destination vectors that can be used in a variety of functional assays. We agree with the reviewer that the word "comprehensive" may be an overstatement given the drop outs and that other genes may also be considered cancer related genes; therefore, we decided to replace "comprehensive" with "large-scale" which is more accurate. We have clarified this in the Results section of the revised manuscript (lines 77-90):

"Systematic studies of TF-DNA binding and transcriptional activity often require large-scale clone resources of regulatory DNA elements - such as promoters and enhancers - that can be tested across functional assays. Considering that promoters are the primary drivers of gene expression while enhancers require more specific spatiotemporal contexts to function effectively, we focused on promoter regions of cancer-related genes to study their regulation by TFs (Bergman et al. 2022). We initially selected 697 genes from the Cancer Gene Census (Tate et al. 2019), as well as 114 additional genes whose expression is often dysregulated in cancer. For 190 of these genes we have also included alternative promoters, in cases where H3K27ac or H3K4me3 marks were observed in data from the ENCODE Project (ENCODE Project Consortium 2012). We successfully cloned 700 promoter sequences - each comprising 2 kb of

sequence immediately upstream of a transcription start site - corresponding to 556 cancer-related genes, generating a Gateway-compatible resource for easy transfer into different destination vectors that can be used in a variety of functional assays (e.g., eY1H, pY1H, and luciferase assays) (Figure 1A, Supplementary Table 1)."

In addition, we provided more details in the methods section (lines 420-431):

"Entry clones and yeast strains for promoters of cancer-related genes were generated following established procedures (Fuxman Bass et al. 2016a; Fuxman Bass et al. 2016b). We selected 697 genes from the Cancer Gene Census (Tate et al. 2019), as well as 114 additional genes whose expression is often dysregulated in cancer. For 190 of these genes, we included alternative promoters in cases where H3K27ac or H3K4me3 marks were observed in data from the ENCODE Project (ENCODE Project Consortium 2012). Promoters encompassing approximately 2 kb upstream of the transcription start site were amplified from human genomic DNA (Clontech) using primers flanked with Gateway tails (Supplementary Table 1, 9). After two rounds of cloning attempts, we successfully cloned 700 promoter sequences corresponding to 556 cancer-related genes into the pDONR-P4P1R vector using BP Clonase (ThermoFisher #11789100), resulting in a collection of Gateway entry clones whose sequences were verified through Sanger sequencing (Sequegen)."

2. Discussion of panels D and E of Figure 1 inverts them.

We have fixed this error in the Figure legend.

3. The use of several pale, low-saturation colors for the edges in Figure 2A makes them indistinguishable to a reader with even partial colorblindness.

We thank the reviewer for the suggestion to improve the clarity of the figure. The revised Figure 2A now uses more distinguishable edge colors.

4. The "111 interactions that were reported in the literature" listed in Supplementary Table 2 should include references (PMIDs would be sufficient).

As suggested, we have added the PMIDs for the corresponding references in Supplementary Table 2.

5. There is surprisingly little discussion of the "631 novel interactions" observed. How many of these represent TFs for which high-quality target identification data were not previously available, versus how many correspond to binding events of TFs that have previously been assayed by genome-wide location methods and not observed to bind there? Is there any other evidence to support the in vivo relevance of these interactions?

We thank the reviewer for the suggestion. Consistent with previous work (25910213), we found that the “novel interactions” without previous ChIP-seq evidence have been assayed by ChIP-seq in fewer experiments than interactions with evidence. This suggests that novel interactions were likely missed in previous studies because the TFs involved are understudied. We have added this analysis as Supplementary Figure 1A and the corresponding discussion in the Results section (lines 127-132): “*Further, consistent with previous studies, we found that TFs whose interactions did not present evidence by ChIP-seq were assayed less frequently than TFs for which ChIP-seq evidence was found (Fuxman Bass et al. 2015)(Supplementary Figure 1A). This suggests that further ChIP-seq datasets are likely to add evidence for the interactions detected by eY1H and pY1H assays, and illustrates the high quality of our cancer TF-DNA network.*”

6. The section on cooperative vs. antagonistic binding is notably light on references. Have any of these TF pairs besides MYC-MAX been explored for non-additive interactions before?

We added more background on the TF pairs’ interactions as well as the corresponding references (lines 233-240): “*TFs from a variety of families are known to bind DNA cooperatively as heterodimers, including NF- κ B (Oeckinghaus and Ghosh 2009), AP-1 (Karin et al. 1997), STATs (Lim and Cao 2006; Delgoffe and Vignali 2013), nuclear receptors, and other bZIP (Rodriguez-Martinez et al. 2017) and bHLH TFs (de Martin et al. 2021). However, the extent to which cooperativity between TFs is involved in cancer gene targeting has not been systematically explored. Furthermore, DNA-binding antagonism between TFs, in which dimerization prevents binding of a TF to certain DNA targets, has not typically been considered as a widespread transcriptional regulatory mechanism.*”

7. The statement in line 383 that “these observations were confirmed” by luciferase assay is overly broad, since it appears to apply to all three disordered regions of ESR1. The preceding paragraph had stated that increasingly large deletions of the N-terminal IDR led to increased DNA binding, but the two promoters shown in Figure 5D,E show significantly decreased and unchanged binding, respectively.

We thank the reviewer for pointing this out and agree that it was not clear which observations were confirmed by luciferase assay. Specifically, luciferase assays confirmed a loss of binding of the Hinge(GS)₂₃ ESR1 construct to the *AFF2* and *NBL1* promoters. To avoid confusion, we have modified figures 5D and 5E to only include results for the empty vector control, wild-type ESR1, Hinge(GS)₂₃ ESR1, and the K252N mutation in the hinge region and have updated the results text as follows:

“*We confirmed the reduced binding of the Hinge(GS)₂₃ ESR1 construct to the *AFF2* and *NBL1* promoters using reporter-based protein-DNA interaction assays in HEK293T cells treated with 100 nM estradiol (Figure 5D-E).*”

8. It would be helpful to mention in the Methods how many replicates were performed for each test and if they were all performed simultaneously on one plate. Similarly, it seems that the phenotypic classes in Figure 5C are scored algorithmically by the DISHA software; if so, it would be helpful to state this explicitly when they're introduced.

We appreciate the suggestion to improve the clarity of our screening and analysis methods. For both eY1H and pY1H assays, each interaction is assessed across adjacent quadruplicate colonies. For pY1H assays, each TF1-TF2 pair strain is compared to corresponding "TF1 only" and "TF2 only" strains located on the same array plate. Interactions from the ESR1 construct eY1H screen were scored manually on a scale from 0 (no signal) to 5 (very strong signal).

We have added the following to our methods section "eY1H screening":

"Each interaction was tested in adjacent quadruplicate colonies. Interactions were considered positive if at least three colonies displayed reporter activity."

We have added the following to our methods section "pY1H screening":

"Each interaction was tested in adjacent quadruplicate colonies. For each TF-pair strain, corresponding TF1 and TF2 single-TF strains are included in the same plate."

We have added the following to our methods section "eY1H screening of ESR1 constructs":

"Each interaction was tested in adjacent quadruplicate colonies and scored manually on a scale from 0 (no reporter signal) to 5 (very strong binding signal)."

In addition we added two Methods sections that further clarify experiments using the ESR1 constructs. These sections are titled:

-Generation of yeast strains expressing ESR1 constructs (lines 596-606)

-Confirmation of eY1H interactions using mammalian luciferase assays (lines 622-644)

9. The Supplementary Data statement should be updated.

We thank the reviewers for pointing this out. We updated our supplementary data statement accordingly.

July 2, 2024

RE: Life Science Alliance Manuscript #LSA-2024-02641-TR

Dr. Juan Fuxman Bass
Boston University
5 Cummington Mall
Boston, Massachusetts 02215

Dear Dr. Fuxman Bass,

Thank you for submitting your revised manuscript entitled "A large-scale cancer-specific protein-DNA interaction network". We would be happy to publish your paper in Life Science Alliance pending final revisions necessary to meet our formatting guidelines.

- please be sure that the authorship listing and order is correct
- please add a callout for Figure 5G to your main manuscript text

LSA now encourages authors to provide a 30-60 second video where the study is briefly explained. We will use these videos on social media to promote the published paper and the presenting author (for examples, see <https://docs.google.com/document/d/1-UWCfbE4pGcDdcgzcmiuJl2XMBJnxKYeqRvLLrLSo8s/edit?usp=sharing>). Corresponding or first-authors are welcome to submit the video. Please submit only one video per manuscript. The video can be emailed to contact@life-science-alliance.org

A. FINAL FILES:

B. MANUSCRIPT ORGANIZATION AND FORMATTING:

****It is Life Science Alliance policy that if requested, original data images must be made available to the editors. Failure to provide original images upon request will result in unavoidable delays in publication. Please ensure that you have access to all original**

data images prior to final submission.**

The license to publish form must be signed before your manuscript can be sent to production. A link to the electronic license to publish form will be available to the corresponding author only. Please take a moment to check your funder requirements.

Sincerely,

Reviewer #2 (Comments to the Authors (Required)):

The authors have addressed my concerns.

July 4, 2024

RE: Life Science Alliance Manuscript #LSA-2024-02641-TRR

Dr. Juan Fuxman Bass
Boston University
5 Cummington Mall
Boston, Massachusetts 02215

Dear Dr. Fuxman Bass,

Thank you for submitting your Resource entitled "A large-scale cancer-specific protein-DNA interaction network". It is a pleasure to let you know that your manuscript is now accepted for publication in Life Science Alliance. Congratulations on this interesting work.

DISTRIBUTION OF MATERIALS:

Again, congratulations on a very nice paper. I hope you found the review process to be constructive and are pleased with how the manuscript was handled editorially. We look forward to future exciting submissions from your lab.

Sincerely,
